# Phylodynamic assessment of intervention strategies for the West African Ebola virus outbreak

Simon Dellicour [1], Guy Baele [1], Gytis Dudas[2], Nuno R. Faria[3], Oliver G. Pybus[3], Marc A. Suchard[4,5,6], Andrew Rambaut [7,8] & Philippe Lemey [1]

Genetic analyses have provided important insights into Ebola virus spread during the recent West African outbreak, but their implications for specific intervention scenarios remain unclear. Here, we address this issue using a collection of phylodynamic approaches. We show that long-distance dispersal events were not crucial for epidemic expansion and that preventing viral lineage movement to any given administrative area would, in most cases, have had little impact. However, major urban areas were critical in attracting and disseminating the virus: preventing viral lineage movement to all three capitals simultaneously would have contained epidemic size to one-third. We also show that announcements of border closures were followed by a significant but transient effect on international virus dispersal. By quantifying the hypothetical impact of different intervention strategies, as well as the impact of barriers on dispersal frequency, our study illustrates how phylodynamic analyses can help to address specific epidemiological and outbreak control questions.

[1] Department of Microbiology and Immunology, Rega Institute, KU Leuven – University of Leuven, Herestraat 49, 3000 Leuven, Belgium. [2] Fred Hutchinson Cancer Research Center, 1100 Fairview Ave N, 98109 Seattle, WA, USA. [3] Department of Zoology, University of Oxford, Oxford OX1 3PS, United Kingdom. [4] Department of Biostatistics, UCLA Fielding School of Public Health, University of California, Los Angeles, CA 90095, USA. [5] Department of Biomathematics, David Geffen School of Medicine at UCLA, University of California, Los Angeles, CA 90095, USA. [6] Department of Human Genetics, David Geffen School of Medicine at UCLA, University of California, Los Angeles, CA 90095, USA. [7] Institute of Evolutionary Biology, University of Edinburgh, King's Buildings, Edinburgh EH9 3FL, UK. [8] Fogarty International Center, National Institutes of Health, Bethesda, MD 20892, USA. Correspondence and requests for materials should be addressed to S.D. (email: simon.dellicour@kuleuven.be)

The recent Ebola virus (EBOV) epidemic in West Africa emerged around the end of 2013 in the prefecture of Guéckédou in Guinea[1] and caused at least 11,310 deaths among 28,616 recorded cases in Guinea, Sierra Leone and Liberia[2]. It has been argued that the West African EBOV epidemic illustrated problems in the early detection of, and rapid response to, infectious disease outbreaks of public health importance[3]. Various reasons may explain the slow initial response to the West African EBOV epidemic, including poor public health infrastructure and local unfamiliarity with Ebola virus disease, as well as a lack of preparedness by the international community. Because efforts to control the epidemic could not rely on vaccination or effective antiviral drugs, the outbreak response focused on standard medical practices (e.g. case identification and isolation), as well as community practices (e.g. sanitary funeral practices)[4]. Mathematical models have been used extensively to study the dynamics of EBOV transmission (e.g. superspreading events[5]), the performance of local containment measures[6–8] and the potential impact of other hypothetical strategies (e.g. the use of rapid diagnostic tests that were not available yet[9]). The impact of air travel restrictions out of the affected region has also been assessed in detail (e.g. Poletto et al.[10]), but long-range interventions within the region, such as border closures, lockdowns and travel restrictions, may be more challenging to investigate. They are however important to consider because, unlike previous EBOV outbreaks that were confined to remote villages, this outbreak occurred in a highly connected region of Africa with large population centres[4,11], spread over multiple countries, without fully coordinated intervention policies[4]. This connectivity is also relevant to local management strategies because the interacting populations do not necessarily implement policies that are coordinated, as was the case for EBOV in Guinea, Sierra Leone and Liberia[4]. Increasing availability of individual-level spatio-temporal mobility data, e.g. mobile phone network data[11], may offer invaluable opportunities to accommodate human connectivity in modelling efforts (e.g. Lau et al.[12]). Pathogen genetic data represent an interesting alternative or complementary data source because it contains information about spatio-temporal spread that can be extracted using phylodynamic approaches. Although pathogen genomes are routinely used for epidemiological reconstructions, opportunities to harness the power of evolutionary approaches to inform intervention strategies are still scarce. Now that genomic surveillance systems can be deployed for real-time pathogen genome sequencing in resource-limited settings[13], it is critical to examine what information relevant to control strategies can be gleaned from pathogen genomes.

Pathogen genome sequencing is also being used to assist with the identification of unknown infection sources and transmission chains, as pathogen genomes contain valuable information that complements contact tracing efforts. In the case of Ebola, Arias et al.[14] demonstrated that rapid outbreak sequencing in locally established sequencing facilities can identify transmission chains linked to sporadic cases. Consequently, it is unsurprising that there have been calls for making pathogen sequence data openly available in outbreak situations[3,15–17]. In addition to identifying specific transmission pathways, pathogen genome analyses can also shed light on the origins, evolution and transmission dynamics of a pathogen during an epidemic[18]. Early in the EBOV epidemic, analyses such as those by Gire et al.[19] demonstrated that the virus entered the human population in late 2013 and crossed from Guinea to Sierra Leone in May 2014 through sustained human-to-human transmission. The EBOV genome data that was generated also stimulated phylodynamic efforts to characterise transmission dynamics early in the epidemic (e.g. superspreading[20]) and to estimate critical epidemiological

parameters, such as the basic reproductive number[21]. Various molecular epidemiological studies subsequently attempted to trace Ebola spread[13,14,22–26] (see Holmes et al.[18] for a detailed overview), marking the beginning of large-scale real-time molecular epidemiology[18]. All these efforts culminated in an impressive collection of over 1600 EBOV genome sequences, corresponding to more than 5% of known cases[27]. These data represent a unique opportunity to learn lessons about the evolutionary and epidemiological dynamics of an Ebola outbreak.

Although Ebola viral genomes were reported across numerous studies focusing on different time periods and/or geographic areas, the collated genetic data cover the entire epidemic exceptionally well, and sampling intensity correlates strongly with the infection burden in different locations throughout the course of the outbreak[27]. This data set motivated a detailed phylogeographic study that identified the patterns and drivers of spatial spread[27]. Specifically, a generalised linear model (GLM) of transition rates between discrete locations in a Bayesian statistical framework was used to test which causal factors might have influenced the spread of the virus at subnational administrative levels (termed districts in Sierra Leone, prefectures in Guinea and counties in Liberia). By considering a range of geographic, administrative, economic, climatic, infrastructural and demographic predictors, this GLM approach provided support for a gravity model of transmission, albeit one that was attenuated by international borders[27]. The gravity model emphasises the impact of population size on viral dispersal and implies that large urban populations acted as sources, reseeding smaller limited epidemics in more outlying locations. Further, the epidemic was generally less likely to spread across international borders, but did so specifically both early on and late in the epidemic, between administrative areas that share such an international border. More detailed spatio-temporal analyses suggested that border attenuation may have resulted from border closures between Guinea, Sierra Leone and Liberia, although their containment effects were limited. Within the three affected countries, viral spread was not always maintained by continuous transmission in each location, but often by repeated introductions into a location, generating small, well-connected, clusters of cases. This dynamical pattern of connectivity characterises a metapopulation, highlighting the need for responsive, mobile and measured interventions.

Here, we extend the phylogeographic analyses of the West African Ebola epidemic in two different ways. First, we examine the implications of EBOV metapopulation dynamics on particular intervention strategies. Specifically, we assess to what extent limiting long-distance spread, or preventing spread to highly populated locations, might have impacted the epidemic. Second, we introduce continuous diffusion models as an alternative phylogeographic framework, which can characterise aspects of the process of Ebola spread that were not captured by the discrete approach employed by Dudas et al.[27]. We quantify important parameters of spatial spread and demonstrate how a posterior predictive simulation procedure can be used to evaluate potential barriers to transmission, specifically, the impact of border closures. These new evolutionary approaches deepen our understanding of the public health implications of EBOV epidemic dynamics and the extent to which viral spread could be curbed by particular intervention strategies.

## Results

**Assessing the impact of hypothetical intervention strategies**. To understand the implications of EBOV metapopulation dynamics during the 2013–2016 epidemic, we first investigate the impact of hypothetical intervention strategies on epidemic size and

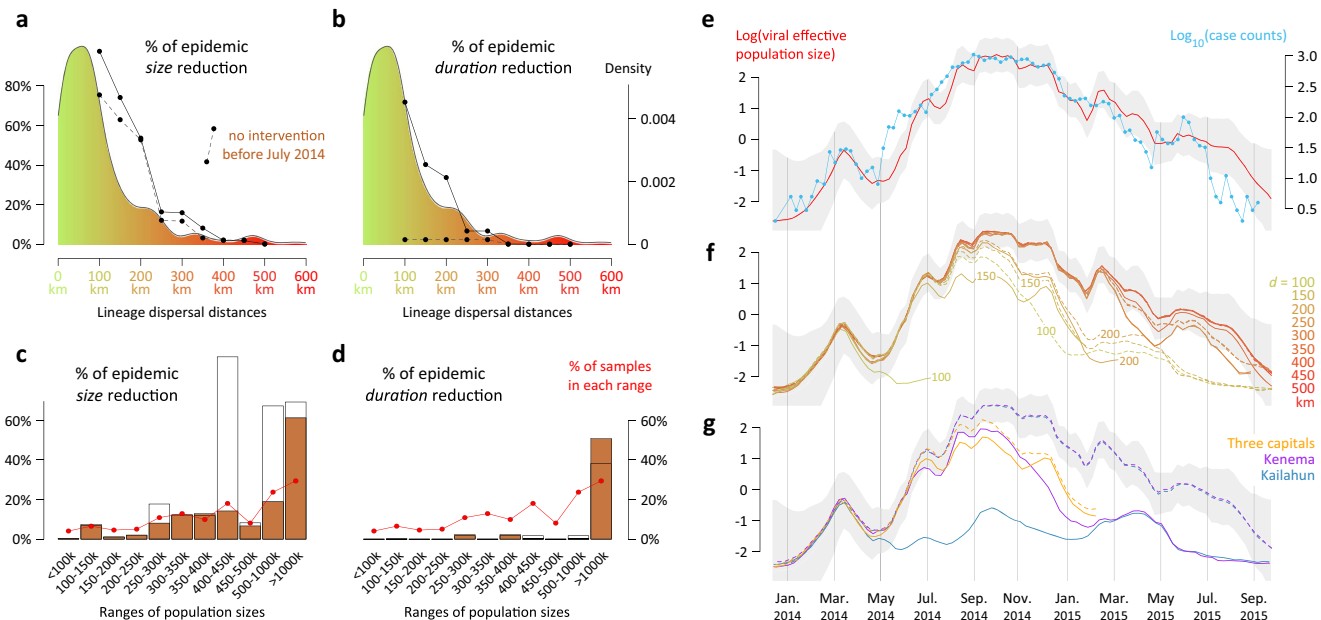

**Fig. 1** Hypothetical impact of intervention strategies. **a**, **b** Reductions in epidemic size and duration following the prevention of dispersal over a range of distances between administrative areas. These quantities are summarised as the percentages by which phylogenetic tree length and height are reduced when the phylogeny is pruned at all the branches that accommodate such dispersal events (full lines). The dashed lines represent the corresponding reductions when the dispersal events are prevented only after June 2014. These curves are superimposed on the distribution of lineage dispersal distances summarised from the posterior Markov jump history (coloured from green to red). **c**, **d** Reductions in epidemic size and duration following the prevention of dispersal to administrative areas belonging to a specific population sizes range. These percentage reductions are also obtained by pruning the phylogeny, but now at all branches that accommodate dispersal events to the relevant administrative areas (white histogram bars). The brown histogram bars represent the corresponding reductions when the dispersal events are prevented only after June 2014. We refer to Supplementary Fig. 1 for credible intervals associated with percentages of epidemic size/duration reductions reported in Fig. 1a–d. All the reductions in tree length and height were computed by conditioning the pruning on movement events recorded in the MCC (maximum clade credibility) tree summary of the discrete phylogeographic reconstruction. Supplementary Fig. 2 summarises the equivalent results for pruning trees using the Markov jump histories associated with each posterior tree. **e** Estimates of viral population size (in red; 95% HPD in grey) and the time series of case counts (in blue). **f** Impact of preventing long-distance dispersal events on viral effective population size through time. As in Fig. 1e, the 95% HPD of viral population size based on the entire dataset (no intervention strategy) is displayed in grey. On this graph, dashed lines correspond to viral population size evolution when transmission is prevented only after June 2014. **g** This plot corresponds to Fig. 1f but focuses on the impact of preventing dispersal events to specific locations on viral effective population size through time

duration. To model the effects of reducing long-distance dispersal (e.g. through travel restrictions), we prune the EBOV phylogenetic trees when a long-distance lineage translocation between administrative areas was inferred, by effectively removing transmission following such dispersal events. We then quantify the effect of this intervention on the epidemic size as reflected in the reduction in tree length in the phylogenetic reconstructions (Fig. 1a). We also report reductions in epidemic duration (based on tree height), but we note that epidemic size is the most relevant measure for the evaluation of the impact of containment strategies. Using this procedure, we assess how important were such long-distance events to the expansion and maintenance of EBOV transmission. For comparison, we also perform an analysis that prevents transmission only after a particular point in time, i.e. by removing only those viral lineage movements that occurred after June 2014. The latter analysis reflects a scenario in which hypothetical intervention strategies are delayed and implemented some time after the onset of the outbreak, in this case after ~6 months, at which time all three countries had already been seeded. For the delayed intervention strategies, percentage reductions in epidemic size and duration are estimated relative to the period of time during which the intervention strategy is effective.

Figure 1a, b depict the impact of preventing long-distance viral lineage movements on relative epidemic size and duration, respectively, under both the immediate and delayed intervention

strategies. Lineage movements between administrative areas involving distances greater than 300 km are rare and the transmission chains they generate do not contribute substantially to the total epidemic size (Fig. 1a). Only if viral lineage movement is impeded over shorter distances (<250 km) do we start to observe a stronger impact on relative epidemic size. This is the case if lineage movements are prevented from the initial stages of the outbreak. If viral lineage movement is restricted after June 2014, the impact on epidemic size only differs for intervention strategies preventing lineage movements over small distances (e.g. a 75% epidemic size reduction at 100 km instead of a 97% reduction without this restriction, Fig. 1a). Epidemic duration is similarly affected if short-distance movements are prevented from the start of the epidemic, but not if the restriction is implemented only after June 2014 (Fig. 1b). This implies that specific viral lineage movement events between administrative areas at least 100 km apart and early in the epidemic were critical for generating long-term transmission chains, and that after June 2014, epidemic duration could have been largely maintained by viral lineage movements over shorter distances, albeit at a smaller epidemic size.

To further investigate the effects of potential interventions on reducing epidemic size through time, we also undertook coalescent inference of viral effective population size through time, which results in estimates that are remarkably proportional to case counts (Fig. 1e). This relationship can be statistically

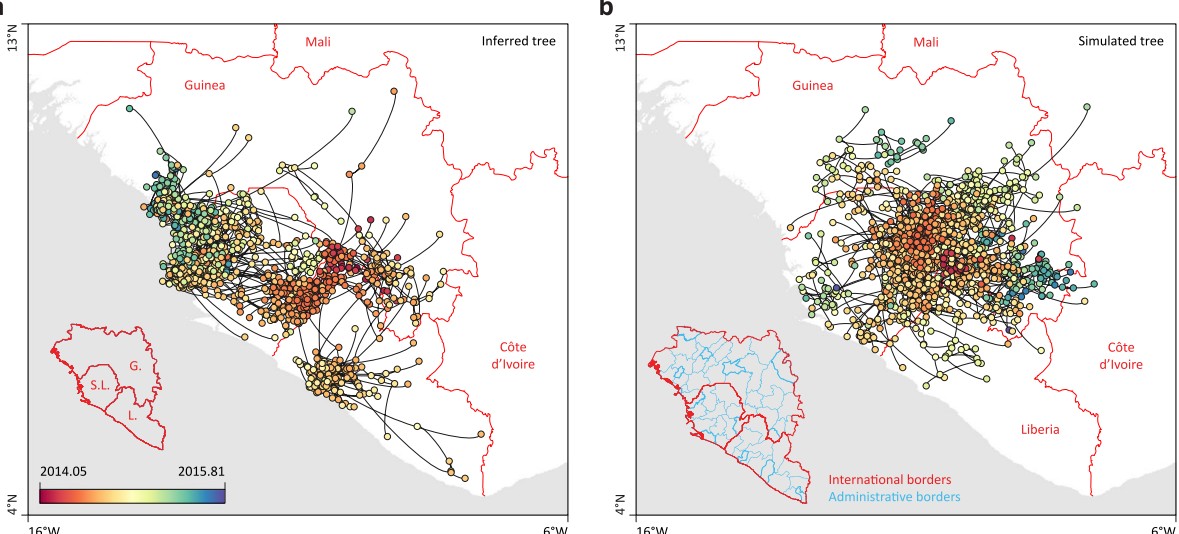

**Fig. 2** Example of a continuous phylogeographic estimate and corresponding simulation. Example of a phylogeographic estimate in continuous space (**a**) and the corresponding posterior predictive simulation unaware of international borders (**b**). In both cases, phylogenetic branches are represented by black curves connecting phylogenetic nodes displayed as dots coloured according to their time of occurrence. In **a**, these nodes are positioned according to the mean latitude and longitude estimates obtained by phylogeographic inference, while in **b** they are positioned according to simulations based on the estimated parameters of the phylogeographic process. Map background was made in R and based on international borders obtained from the Global Administrative Areas database (GADM, www.gadm.org)

tested by using a GLM-based extension of the coalescent approach that considers case counts as a potential covariate of viral effective population size[28]. The GLM coefficient for the association between case counts and estimated effective population size is high (0.55) and its credible intervals exclude zero (95% highest posterior density interval (HPD): 0.18–0.90), indicating a significant association. The effective population size estimates indicate that the impact of pruning dispersal events between administrative areas ≥200 km apart from the phylogenetic trees has a notable effect on epidemic size only after the time of the epidemic peak (Fig. 1f). These estimates also illustrate the pronounced effect on epidemic size and duration of applying a delay to the prevention of viral lineage movement over smaller distances. If viral lineage movement had been prevented over distances >100 km between administrative areas at the onset of the outbreak, then the epidemic would have been restricted to an initial small peak in epidemic size (Fig. 1f), which represents the emergence in the Guéckédou prefecture and neighbouring areas[27].

We next apply a similar procedure, but this time restricting viral lineage movement according to the population sizes of the 'destinations' (administrative areas) of viral lineage movement. Specifically, we bin areas according to their population size and remove all descendent transmission (subtrees) that occur after lineage dispersal events to corresponding areas. We then examine the effect of this restriction on epidemic size and duration assuming, as in the distance-based pruning, that the restriction results from a 100% effective intervention. Figure 1c, d summarise the relative reduction in epidemic size and duration, with and without a delay in transmission prevention (filled and open bars, respectively), and summarise the sample sizes from all areas within each population size range. For two population size ranges (400–450 k and 500–1000 k), the presence or absence of a delay on transmission prevention has a large effect on the observed reduction in epidemic size. By examining the impact of each administrative area separately (Supplementary Fig. 3), we can attribute this difference to the impact of preventing viral lineage movement to the Kailahun and Kenema districts in Sierra Leone

prior to June 2014. These administrative areas represent early key spatio-temporal foci of EBOV dissemination that are specific to this epidemic. The virus spread extremely rapidly from Kailahun district to several counties of Liberia[24] and Guinea[22,26]. However, preventing early viral lineage movement to Kailahun district would not have noticeably reduced the duration of the epidemic (Fig. 1g), because a basal phylogenetic lineage specific to Guinea would have remained unaffected by this restriction and would have continued to circulate, albeit with a limited epidemic burden. We acknowledge that in the interpretation of intervention scenarios, we not only assume that they are 100% effective, but also that people for which infection was prevented in particular areas would not have been infected through other introductions, and that all other efforts would have remained unchanged. With respect to the latter however, a localised lineage causing limited cases in Guinea as a consequence of interventions would have been more easily contained. Preventing early spread to Kenema district would also have halted much of the westward spread from Kailahun district, but its impact on epidemic size is smaller (Fig. 1g), because Kenema district was much less important than Kailahun district for viral spread to Liberia. From these two districts, EBOV disseminated over relatively long distances to other administrative areas, which is why preventing early lineage movements over such distances generates a strong reduction in epidemic size in the distance-based examination.

When applying a delay to intervention strategies, the effect of preventing viral lineage movement to sets of administrative areas with different population sizes generally had a limited impact on predicted epidemic size (Fig. 1c) and virtually no impact on epidemic duration, except for the case where viral lineage movement was prevented to areas with population sizes >1,000,000. This category corresponds precisely to the areas encompassed by the three capitals, i.e. Montserrado, Freetown (and suburbs) and Conakry. Although about 28% of the genome samples were from these administrative areas (and about 39% of reported cases), removal of viral lineages that moved into these areas lead to a disproportionate reduction in epidemic size, of about 60%, which starts to take effect before the epidemic peak

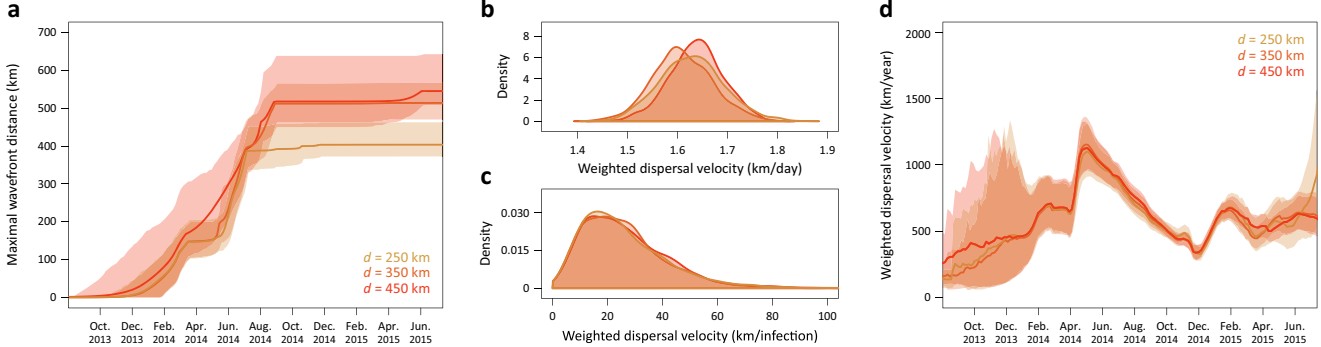

**Fig. 3** Dispersal statistics summarised from continuous phylogeographic inference. Evolution of maximal wavefront distance (**a**), mean dispersal velocity (**b**, **c**) and dispersal velocity through time (**d**) for each data set ($d = 250$, 350 and 450 km). These statistics were summarised from 1000 trees sampled from the posterior distribution of trees. Weighted dispersal velocities are reported in both km per day (**b**) and in km per infection (**c**). Weighted dispersal velocities in km per infection are obtained by multiplying mean branch velocities (in km/day) by serial interval values (in days between successive infections) randomly drawn from the generation time distribution estimated by the WHO Ebola Response Team (2014)

(Fig. 1g). This reflects the important role of highly populated locations in fuelling EBOV transmission, as previously highlighted[27]. We note that the epidemic size reduction for the three capitals together is less than the sum of the reduction obtained by removing viral lineage movement to each capital individually (Supplementary Fig. 3), indicating that there was transmission among the capitals. Furthermore, we observe a substantial reduction in epidemic duration when preventing the viral lineage movement to all three capitals (i.e. administrative areas with >1000 k people; Fig. 1d), but not when preventing viral lineage movement to a single capital (Supplementary Fig. 3). However, this result appears to be sensitive to conditioning the tree pruning on the maximum clade credibility (MCC) tree with its annotated movement events. The MCC tree is one of the few trees in the posterior distribution for which a capital is important for the maintenance of all residual lineages in the late stage of the epidemic (see the comparison between Supplementary Figs. 1 and 2). So, epidemic duration is not only less relevant for assessing intervention strategies, it is also less robust to the phylogenetic procedure we employ.

**Phylogeographic estimates in continuous space.** In the second part of this study, we use a new continuous phylogeographic approach to further address other aspects of EBOV spatial spread. Because human mobility can make pathogen diffusion highly irregular when measured against geographic distance, viral spread may not always be adequately modelled by a Brownian-like diffusion process[29,30]. However, the strongly distance-dependent diffusion of the EBOV (Fig. 1a) justifies its use on this relatively restricted geographic scale. Here we employ a relaxed random walk (RRW) model that accommodates diffusion rate heterogeneity among lineages. We assess the sensitivity of our continuous phylogeographic reconstructions to long-distance dispersal by analysing the data sets that were pruned based on discrete transitions larger than 450, 350 and 250 km (see above). Figure 2a illustrates an EBOV phylogenetic tree estimated under this model of continuous phylogeographic diffusion, which is spatially mapped onto the study area.

Our phylogeographic estimates of the epidemic wavefront through time indicate that EBOV spread up to ~500 km from its location of origin in about 8–9 months (Fig. 3a). With a maximum wavefront distance of ~400 km, the same extent of spatial spread is not achieved for the data set restricted to dispersal events <250 km, indicating that relatively long-distance dispersal events contributed to the maximum epidemic wavefront distance. The velocity of the epidemic wavefront from the

beginning of 2014 to early September 2014 is ~1.9 km/day, which is consistent with, but still smaller than the estimated velocity of spread of 2.8 km/day based on weekly counts of confirmed cases per administrative area[31]. Our continuous phylogeographic approach also estimates a mean dispersal velocity of 1.64 km/day (95% HPD [1.52, 1.74]), with little variability across data sets (Fig. 3b). Based on a mean serial interval of transmission of 15.3 days and its uncertainty (SD = 9.3 days[32]), this translates to a mean dispersal distance of 25.4 km per infection (95% HPD [5.33, 61.21]; Fig. 3c) confirming that transmission was fuelled by relatively high mobility of infected individuals in this region of Africa[11]. Our estimates are mostly informed by dispersal between districts, prefectures and counties (see Methods section), and therefore ignores a lot of local transmission within these administrative areas. However, similar analysis of an alternative data set of genomes from Sierra Leone with more precise coordinates and multiple samples per administrative area provided largely consistent estimates (see Supplementary Note 1). The mean dispersal velocity shows a remarkable variability during the course of the epidemic (Fig. 3d). It steeply increases from January 2014 to a peak around May/June, coinciding with the time of spread across the region from Kailahun. The peak in mean dispersal velocity is followed by a marked drop until the end of the year 2014. This drop appears to begin before the announced border closures (Sierra Leone on 11 June 2014, Liberia on 27 July 2014 and Guinea on 9 August 2014). In addition, we observe the same pattern even if we summarise dispersal velocity only for branches that do not cross-borders (as determined by their inferred node locations), suggesting that the decrease in dispersal velocity may be attributed to the impact of a more general awareness of the outbreak, and of the emerging response against it.

**Impact of borders on dispersal frequency.** Although the mean dispersal velocity may be affected by factors other than border closures, analyses using the discrete phylogeographic approach showed that the frequency at which national borders were crossed by viral lineages significantly decreased following the announced border closures between the countries[27]. In order to evaluate this hypothesis using our continuous phylogeographic approach, we developed a test procedure that compares the estimated posterior frequency of border crossing events to the same frequency in posterior predictive simulations that are unaware of borders (see the Methods section, as well as Fig. 2b for an example of such simulation). We quantify the deviation in estimated border frequency crossing from the expected frequency obtained by

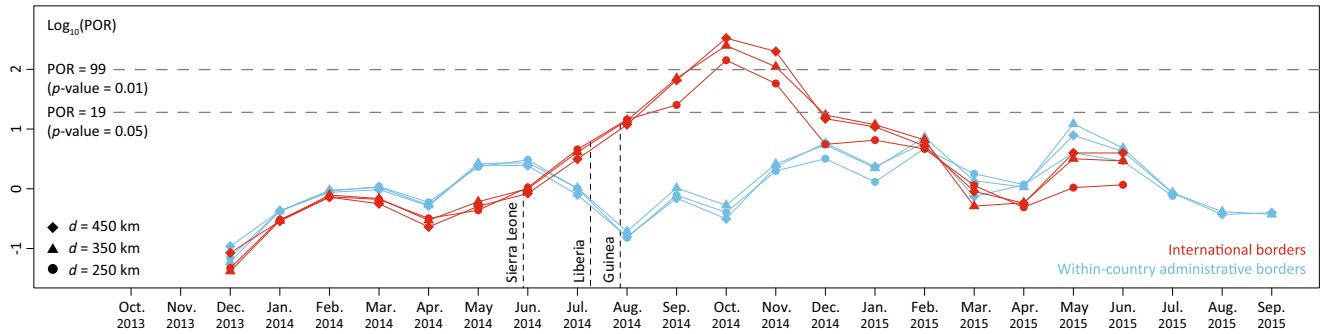

**Fig. 4** Analysis of the monthly impact of inter- and intra-national borders on the EBOV dispersal frequency. The plot depicts the predictive odds ratio (POR) estimates per month and per data set ($d = 250$, 350 and 450 km). POR estimates for within-country administrative borders are also included as a negative control as we do not expect any significant impact of within-country administrative borders on dispersal frequency. PORs >3 and >20 can be considered as 'positive' and 'strong' evidence for the impact of borders on the dispersal frequency, and PORs >19 and >99 correspond to posterior predictive $p$-values <0.05 and <0.01, respectively. Vertical-dashed lines indicate the time at which Sierra Leone, Liberia and Guinea announced their border closures. See Supplementary Fig. 4 for a more detailed representation of the monthly differences in crossing border events between inferred and simulated diffusion processes

posterior simulation as a predictive odds ratio (POR), and calculate posterior predictive $p$-values to assess statistical significance. We also perform this analysis using the data sets for $d = 250$, 350 and 450 km and include the frequency of within-country administrative border crossing as a negative control (because we do not expect any significant changes in this frequency as a result of border closures). In Fig. 4, we plot the POR estimates per month (see also Supplementary Fig. 4 for a more detailed representation of the monthly differences in crossing border events among inferred and simulated diffusion processes). These estimates provide strong evidence for a significantly reduced frequency of international border crossing from September 2014 (posterior predictive $p$-values < 0.05), i.e. starting shortly after the announced border closures (Fig. 4), to about the end of the year. As expected, the frequency of within-country administrative border crossing does not depart significantly from that produced by simulations that are unaware of these borders.

## Discussion

West Africa has experienced the largest outbreak of EBOV in history, with more cases and fatalities than all reported outbreaks combined since 1976. Although the region has been declared Ebola-free since 2016, it remains critically important to learn as much as possible from this devastating epidemic. In the first part of this study, we investigate the two key elements of a gravity model of spread, distance and population size, by measuring the predicted reduction in epidemic size (and duration) that results from restricting viral lineage movement, either by preventing viral lineage movements over varying distances, or by preventing movement to areas of different population sizes. The latter can be used to evaluate the impact of hypothetical intervention strategies. We found that long-distance dispersal events were not critical for epidemic expansion, and only when dispersal events are restricted to 200 km or less did we observe significant reductions in epidemic size. While this result does not immediately translate into practical intervention strategies, it suggests that frequent short-distance dispersal may be more important than rarer long-distance dispersal events in driving the epidemic spread.

We demonstrate that the contribution of population size to the previously identified gravity model of EBOV spread[27] is primarily driven by viral lineage movement to and from the areas encompassing the three capital cities, which are the most highly populated areas in the outbreak region. The fact that the West African EBOV epidemic also affected urban areas, in addition to rural areas, makes it stand apart from all previous EBOV

outbreaks. If viral lineage movement to a single capital could have been prevented, beginning from the onset of the epidemic, then epidemic size could have been reduced by 15–37%. Preventing lineage movement to all three capitals would have reduced epidemic size by two-thirds, while their sample size percentage and case count percentage are 28% and 39%, respectively. This result emphasises the importance of urban transmission, but at the same time, it indicates that no single capital was critical for the maintenance of all co-circulating lineages. The inability to strongly reduce epidemic size by preventing viral lineage movement to other collections of locations further underscores the highly pervasive and distributed nature of the metapopulation dynamics underlying the epidemic. Together with the metapopulation dynamics highlighted by Dudas et al.[27], the picture that emerges from phylogeographic analyses is one of multiple moving targets: potential intervention strategies that are piecemeal, reactive and geographically restricted are predicted to have a limited impact on epidemic size and duration. These dynamics argue for coordinated intervention strategies across the whole outbreak region. By applying a delay to the hypothetical intervention strategies, which is a realistic scenario for most outbreaks, we avoided the impact of preventing early transmission from Kailahun district and Kenema district in Sierra Leone. The importance of these early dissemination centres during the 2014–2016 EBOV outbreak, and their role in spreading the virus to other areas—in particular for Kailahun district—has been highlighted before[22,24,26].

Our phylogenetic approach of assessing hypothetical containment strategies rests on a number of assumptions, with a 100% effectiveness of their implementation being an important one. While it would be straightforward to introduce a probability on the effectiveness of preventing the movement events we target, quantifying the corresponding impact using our phylogenetic measures may not be so relevant. Even if only a fraction of movements is allowed to escape prevention, the resulting transmission chains in the relevant area may have put everyone at risk of infection. In other words, our approach needs to assume that persons that were not infected by a particular lineage, because its transmission was halted, were not exposed to other transmission chains that were not contained. Our phylodynamic approach therefore offers a best-case scenario as a starting point; different degrees of effectiveness and their potential nonlinear impacts on outcomes may be examined in future applications of computational models. Further investigations will be important to assess whether interventions, such as travel restrictions, can in practice

be implemented with reasonable success. In the case of air travel and influenza spread for example, travel restrictions were shown to be largely unable to effectively contain the international spread of a pandemic[33].

Phylogeographic reconstructions in continuous space have been used primarily for animal viruses because the dispersal of their hosts may be reasonably approximated by a RRW process[34–38]. However, the distance-dependent transmission dynamics of EBOV also justify the use of a continuous diffusion process for the phylogeographic reconstruction of this outbreak. We present this as an alternative and complementary approach for the study of spatial epidemiological dynamics in human populations, at least for well-sampled outbreaks at relatively restricted geographic scales. Continuous phylogeographic reconstructions enable us to quantify several aspects of dispersal dynamics, such as mean dispersal velocity per infection (25.4 km per infection [5.33, 61.21]). We observe a strong heterogeneity in this mean dispersal velocity over time, with a significant decrease from May/June 2014 until the end of that year. This likely reflects the general impact of control strategies and awareness on human behaviour.

Several of our findings are in line with or are complementary to the results of modelling studies. Based on case occurrence data, Backer et al.[39] estimated that only 4–10% of newly infected EBOV cases migrated to another district and that, among these migrants, only 0–23% left their country of origin. Kramer et al.[40] used a spatial network approach to demonstrate that the probability of viral lineage movement between locations depended on international border closures. Phylogeographic approaches can contribute important insights by directly inferring the historical connections underlying viral spread. In our study, we use posterior predictive simulation in a continuous phylogeographic framework to assess changes in international border crossing through time. Our findings confirm a significant decline following the announcements of border closures between Guinea, Sierra Leone and Liberia, which was previously observed using a discrete phylogeographic approach[27]. The procedure we used here has a number of advantages relative to the discrete phylogeographic approach that uses epoch modelling to incorporate time-varying predictors[27,41]. A continuous phylogeographic reconstruction does not require a prior specification of the number of change-points, which, in the previous EBOV analysis, was restricted to a single change-point[27]. Instead, we flexibly identify the relevant time period by using a statistic that deviates from the null expectation. In addition, we avoid a discrete approach with large-state spaces that arise from large numbers of locations and that are very time consuming to compute, despite the ability to employ multicore GPU architecture[42].

Our analysis underlines how border closure decisions may spatially structure an epidemic without necessarily having a strong containment effect. The methodology presented here could be used to study the impact of potential barriers on the epidemic spread of other important pathogens. Assessing the impact of hypothesised intervention strategies, as well as the impact of border closures is of interest for public health agencies and policy makers and may provide a better general understanding of outbreak dynamics.

## Methods

**Assessing the impact of hypothetical intervention strategies.** Similar to Ratmann et al.[43], we use a phylogenetic pruning approach to investigate prevention strategies, but now specifically relying on the associated estimates of spatial spread. We build on the phylogeographic reconstruction performed by Dudas et al.[27], who used a GLM-parameterisation of discrete phylogenetic diffusion[44]. Based on a data set of 1610 viral genomes sampled between 17 March 2014 and 24 October 2015 (available at https://github.com/ebov/space-time), Dudas et al.[27] used this approach to reconstruct a history of lineage movements between 56 administrative regions in Guinea (prefectures), Sierra Leone (districts) and Liberia (counties). Their Bayesian

inference resulted in a posterior distribution of time-measured trees, each annotated with inferred ancestral locations, which was summarised as an MCC tree. In order to assess the impact of preventing viral lineage movement over specific geographic distances, or to specific locations, we condition on the full transition history in the MCC tree recorded using Markov jumps. Markov jump estimation provides a stochastic mapping of the realisations of the continuous-time Markov process throughout evolutionary history[45,46]. Because each lineage movement between a pair of locations is associated with a geographic distance (the great-circle distance between the locations' population centroids), we are able to assess the impact of preventing viral lineage movement over distances $>d$ by pruning from the complete tree all subtrees that represent the transmission history following branches that accommodate such lineage movement. As possible values for $d$, we test a series of decreasing distances: 500, 450, 400, 350, 300, 250, 200, 150 and 100 km; this yields pruned trees with 1607, 1567, 1567, 1498, 1368, 1242, 875, 383 and 53 sequences, respectively. In order to prune the same taxa from all posterior trees (using PAUP*[47]), such that pruned trees can also be used as empirical tree distributions in subsequent coalescent inference (cfr. below), we condition on the Markov jump history in the MCC tree to determine which taxa need to be pruned. However, for the measures of epidemic size and duration, we examine the sensitivity to conditioning on the MCC tree by also pruning each tree of the posterior distribution based on its specific Markov jump history (Supplementary Fig. 2). From the resulting set of pruned posterior trees, we use the program TreeStats[48] to compute the tree length (the sum of all branch lengths) and tree height (the time to the most recent common ancestor), which we interpret as measures of relative epidemic size and epidemic duration. In order to also obtain an estimate of the effect on relative epidemic size through time, we make use of coalescent estimates under a flexible Bayesian skygrid model[49]. For the complete data set, the effective population size estimates are highly correlated with case counts through time (Fig. 1a). Conditioning on a posterior subset of 1000 pruned genealogies, we re-estimate effective population sizes through time using the Bayesian skygrid model and compare these estimates to the original coalescent estimates. All the effective population size plots are based on summaries from the program Tracer 1.7.

We follow a similar procedure to assess the impact of preventing viral lineage movement to a specific category of administrative areas or to individual administrative areas. Different categories of administrative areas were defined on the basis of their population size: <100, 100–150, 150–200, 200–250, 250–300, 300–350, 350–400, 400–450, 450–500, 500–1000 and >1000 k people. In these analyses, we pruned subtrees from the complete tree that were the result of movements from any location to the location(s) within the category under consideration. As before, we obtain the estimates of epidemic duration, relative size, and relative size through time.

**Bayesian skygrid estimation with covariates.** To assess the strength of association between case counts and effective population size through time, we use a recent extension of the non-parametric Bayesian skygrid model that incorporates potential covariates[28]. This approach allows us to include external time series as covariates in a GLM framework while accounting for demographic uncertainty. By applying this GLM framework to the complete genome data set[27], we model the Ebola outbreak effective population size as a log-linear function of case counts and estimate the effect sizes for the latter as a GLM coefficient.

**Continuous phylogeographic inference.** As an alternative to discrete phylogeographic inference, we estimate the spatio-temporal dynamics of EBOV by inferring viral lineage movements in continuous space with a multivariate diffusion approach implemented in BEAST[50]. This approach infers ancestral locations (in geographic coordinates) of internal nodes using a (relaxed) random walk diffusion process. In order to assess the impact of long-distance dispersal on the continuous diffusion estimates, we analyse the pruned data sets obtained using the procedure described in the previous section. Specifically, we remove the same taxa as in the subtrees that need to be pruned based on viral lineage movements with distances $>d$ on particular branches. For computational convenience, we restrict ourselves to specific values of $d$, i.e. $d = 450, 350$ and 250 km. For most sequences, only admin-2 level locations of sampling are known (administrative areas that correspond to prefectures in Guinea, districts in Sierra Leone and counties in Liberia). We therefore remove sequences without a known administrative region of sampling (e.g. when only the country of sampling is known), and we remove sequences such that monophyletic clusters of sequences sampled from the same administrative region are only represented by a single sequence. Such clusters would largely represent dispersal within administrative regions, which will be characterised by 'noise' because of their randomly drawn geographic coordinates within the administrative region (see below). The final data sets include 722 ($d = 450$ km), 676 ($d = 350$ km) and 527 ($d = 250$ km) sequences, respectively.

Inference under the multivariate diffusion models is problematic when different sequences are associated with identical trait values[50], or in this case, with the same geographic coordinates. When unique sampling coordinates are not available for every sequence, a common practice is to add a restricted amount of noise to duplicated traits. In BEAST[48], this is facilitated by a jitter option, that uniformly draws such noise from a user-defined window size for each dimension of the trait (i.e. each coordinate). Using a jitter may be problematic for two reasons in our case. Many administrative areas are along the coast, so the added noise may lead to

sampling coordinates in the sea. Due to the different sizes and shapes of the administrative areas, the noise may also move coordinates to areas neighbouring their actual sampling area. To avoid these issues, we associate a random coordinate within the administrative area of sampling to each sequence. Because this approach ignores a lot of short-distance transmission within administrative areas, we show that our estimates are consistent with those based on an analysis of a data set that is composed of sequences with more precise sampling locations (the admin-3 chiefdom level) and that does not restrict monophyletic clusters of sequences sampled from the same administrative region to a single representative. The details of this additional analysis and the associated results are reported in Supplementary Note 1.

We analyse each data set using a RRW model with an underlying Cauchy distribution to represent among-branch heterogeneity in branch velocity[50], and with a multivariate Wishart distribution as a prior on the precision matrix[51]. We follow Dudas et al.[27] in choosing substitution, molecular clock and coalescent models, and their prior specifications, and reiterate those choices here. We model molecular evolution according to a HKY + $\Gamma_4$[52,53] substitution model independently across four partitions (codon positions 1, 2, 3 and non-coding intergenic regions), allowing for partition-specific relative rates. We use a non-parametric coalescent Bayesian skygrid model as a prior density over the tree[28,49] and model branch-specific evolutionary rates according to a relaxed molecular clock with an underlying log-normal distribution[54]. We specify a continuous-time Markov chain reference prior[55] for the overall evolutionary rate, while the rate multipliers for each partition were given an uninformative uniform prior over their bounds. All other priors in the phylogenetic inference are left at their default values. For each of the continuous phylogeographic inferences, we ran an MCMC chain using BEAST 1.8.4[48] for 130 ($d = 250$ km), 470 ($d = 350$ km) and 740 ($d = 450$ km) million iterations, removing the first 5% samples in each chain as burn-in. Based on these continuous phylogeographic inferences, we estimate, for each data set, the mean or weighted dispersal velocity $v_{weighted}$ defined as follows:

$$v_{weighted} = \frac{\sum_{i=1}^{n} d_i}{\sum_{i=1}^{n} t_i}$$

where $n$ is the number of branches in the phylogeny, $d_i$ the geographic distance travelled and $t_i$ the time elapsed on each phylogeny branch. We estimate this statistic for 1000 samples of the posterior distribution of trees and report a posterior distribution of estimated values. In addition, we also summarise, for each data set, the evolution of mean dispersal velocity and of the wavefront distance through time[34]. For the latter, we plot the distance between the estimated location at the root and the lineage that is estimated to be the furthest from the root location, summarised for a series of time-slices of the posterior tree distribution. We obtain all these statistics using the R package 'seraphim'[56,57].

**Impact of borders on dispersal frequency**. To test if administrative/international borders act as barriers to dispersal frequency, we adopt a Bayesian posterior predictive simulation procedure. Such a procedure allows the calculation of a Bayesian counterpart of the classical p-value, using posterior predictive replications of the data and employing a test statistic that depends on both data and unknown (nuisance) parameters[58,59]. In our setup, we record the number of times borders are crossed by tree branches in the posterior set of trees, as determined by the location at the parent and child node of the branches, and compare this to posterior predictive values for the same statistic. We obtain the posterior predictive values by simulating a forward-in-time RRW process along each posterior tree using the sampled precision matrix parameters and location at the root node. We also condition on the branch-specific posterior rate scalars to generate the mixture of normals that characterises the RRW. In addition, we constrain the RRW simulations such that the simulated node locations remain within continental Africa (and do not fall in the ocean). Figure 2 illustrates the difference in position and orientation of branches between a diffusion history reconstructed from the data and a diffusion history simulated using our posterior predictive simulation procedure. We integrate over all possible realisations, weighted by their posterior probabilities based on 1000 samples, to generate a test based on the border crossing frequency. In the absence of a border impact on Ebola movement in the West African epidemic, we expect a similar number of border crossing events between the posterior and posterior predictive diffusion histories, as the latter is deliberately unaware of borders. For each data set, we again use a sample of 1000 trees from the post burn-in posterior distribution of phylogenies in order to accommodate phylogenetic uncertainty. Upon simulation of RRW diffusion along these sampled trees, we count and compare the number of crossing border events for each pair of inferred and simulated diffusion processes associated with a particular tree. Each 'inferred' N value ($N_{inferred}$) is thus compared to its corresponding 'simulated' value ($N_{simulated}$) to compute a POR as follows:

$$POR = \frac{p_e}{1 - p_e} \bigg/ \frac{0.5}{1 - 0.5}$$

where $p_e$ is the posterior probability that $N_{inferred} < N_{simulated}$, i.e. the frequency at which $N_{inferred} < N_{simulated}$ in the sampled posterior distribution. In interpreting POR estimates, we adopt the Bayes factor scale defined by Kass & Raftery[60]: values higher than 3, 20 and 150 are considered as 'positive,' 'strong' and 'very strong'

evidence respectively for the impact of administrative/international borders on dispersal frequency. Both types of within-country administrative borders were obtained from the Global Administrative Areas database (GADM, www.gadm.org).

**Code availability**. The posterior predictive simulation procedure is implemented, along with a related tutorial, in the R package 'seraphim'[56,57].

**Data availability**. The BEAST XML files of the new continuous phylogeographic analyses are available at https://github.com/ebov/space-time. The authors declare that all other data supporting the findings of this study are available within the article and its Supplementary Information files, or are available from the authors upon request.

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

## Acknowledgements

S.D. is a postdoctoral research fellow funded by the Fonds Wetenschappelijk Onderzoek (FWO, Belgium) and by the Fonds National de la Recherche Scientifique (FNRS, Belgium). P.L., G.B. and A.R. acknowledge funding from the European Community's Seventh Framework Programme (FP7/2007–2013) under Grant Agreement no. 278433-PREDEMICS. P.L., A.R. and M.A.S. acknowledge funding from the European Research Council under the European Union's Horizon 2020 research and innovation programme (grant agreement no. 725422-ReservoirDOCS) and from the Wellcome Trust Collaborative Award, 206298/Z/17/Z. O.G.P. is supported by the European Union's Seventh Framework Programme (FP7/2007-2013)/European Research Council (614725-PATH-PHYLODYN) and by the Oxford Martin School Programme on Pandemic Genomics. P. L. acknowledges support by the Special Research Fund, KU Leuven ('Bijzonder Onderzoeksfonds,' KU Leuven, OT/14/115), and the Research Foundation–Flanders ('Fonds voor Wetenschappelijk Onderzoek–Vlaanderen,' G066215N, G0D5117N and G0B9317N). N.R.F. is funded by a Sir Henry Dale Fellowship (grant 204311/Z/16/Z). M. A.S. acknowledges funding through the National Science Foundation grant no. DMS1264153 and National Institutes of Health grant no. R01 LM012080. G.B. acknowledges support from the Interne Fondsen KU Leuven/Internal Funds KU Leuven. G.D. is supported by the Mahan Postdoctoral Fellowship at Fred Hutchinson Cancer Research Center. The VIROGENESIS project receives funding from the European Union's Horizon 2020 research and innovation program under grant agreement no. 634650. The computational resources and services used in this work were provided by the VSC (Flemish Supercomputer Center), funded by the Research Foundation—Flanders (FWO) and the Flemish Government—department EWI.

## Author contributions

S.D. and P.L. designed the study. S.D., G.B., A.R., G.D. and P.L. performed the analysis. S. D. and P.L. wrote the manuscript. M.A.S. provided statistical guidance. All authors discussed the results, edited and approved the contents of the manuscript.

## Additional information

**Competing interests:** The authors declare no competing interests.



