## [Peer Review File · Nature Communications]

Reviewers' comments:

Reviewer #1 (Remarks to the Author):

This paper gives an account of a suite of phylodynamic/phylogeographic analyses of the recent West African Ebola outbreak. Although the sequences analysed have been published previously, by building on the results described in the previous paper this work generates important insights into the factors underpinning the outbreak. Specifically, the authors have used advanced phylogenetic approaches to evaluate the impact of long distance transmission, transmission to locations of defined population sizes, and the impact of border closures on the Ebola epidemic, as well as characterising the spatial spread of the virus. I find the paper to be well written, and interesting in both its methodological approaches and applied results. I feel the methods are appropriate and valid, and presented in sufficient detail to be reproducible. However, I do have some comments and queries on the manuscript which I feel should to be addressed prior to publication:

Abstract

1. L9: Please clarify what is meant by "single locations".
2. L11: Please clarify this sentence with regards "'transit centres' for transmission chains".

Introduction

3. Although the introduction provides useful background to the general literature surrounding the Ebola outbreak, it could be improved by refining the section to more specifically frame the work presented here and the questions it answers. This especially relates to the first paragraph of the Introduction.

Results

4. L102-103: please could you be more clear about what the 97%/75% figure relates to, and ideally rewrite the sentence more clearly.
5. L197: Is "cfr" a typo?

Materials and methods

6. L290 on: How well supported were the ancestral traits for the nodes of the MCC tree? If nodes with poor posterior trait support were present, how did were they accounted for in your analysis?
7. L290 on: Can you comment on whether pruning based on the subtrees identified from the MCC tree (rather than identifying the relevant subtrees individually for each of the posterior trees in turn) is likely to have influenced your results, or the confidence reported for your results?
8. L305-306: Can you comment on your rationale for using exclusive population size intervals here (e.g. 100-150k, 150-200k), as compared to the inclusive categories used for the distance-based pruning (L292-293).
9. Paragraph L330 on:
 - 9a. I feel the location uncertainty in the data with respect to this part of the analysis should be better highlighted in the Results (e.g. at L168) and/or Discussion (e.g. L247-251).
 - 9b. Please provide a map of the administrative regions concerned in the supplement, so that

the reader can get a better idea of the spatial resolution/uncertainty present in the data.

9c. Were the randomised location coordinates drawn separately at the start of each BEAST run, and, if so, did this lead to any differences in the estimates resulting from each independent run (if you ran multiple independent chains for each distance class, see point below)?

10. L350: Does this line indicate you ran only a single MCMC chain for each of the distance classes? If so, is there any concern over whether the single set of randomly-chosen starting locations are likely to affect the results for each of these distance classes (see the previous point)?

11. L369: Change "African" to "Africa"

Figures

12. Figure 1A and B: Please described in the figure legend the meaning of each of the lines (solid and dashed) and the shaded distribution. Additionally, it would be good to remove the (solid grey-ish) line from the edge of the distribution for clarity.

13. Figure 1 C and D: No sign of the dashed line described in the legend. Please also indicate the meaning of the shaded and white bars.

14. Figure 1 E: Might be best to change the colours to avoid green and red, in case of colour-blindness.

15. Figure 1 F contains a lot of information: is there any way of changing the figure to make this information any clearer or more accessible to the reader?

16. Figure 4: Please indicate in the legend the meaning of the different colours. Again, please try and avoid green and red (for the colour-blind).

Reviewer #2 (Remarks to the Author):

"Phylogenetic assessment of intervention strategies for West African Ebola virus outbreak" by Dellicour et al. presents some very interesting insight into the implications of different intervention scenarios on the spread and size of the Ebola virus outbreak. The authors use phylogenetic approaches to explore different intervention strategies. They show that 1) long distance dispersal events did not have a large effect on the epidemic expansion, 2) that the three capitals were hubs of transmission and isolating these at the same time would only have reduced the epidemic size by one third, 3) that the dispersal velocity increased steeply from January 2014 to June and decreased markedly following border closures in July 2014 and this pattern was still present in the absence of branches crossing borders suggesting that the decrease in dispersal velocity could be attributed to general awareness of the outbreak rather than border closure.

This manuscript is novel and of broad importance to the research community, additionally it is likely to influence future Ebola virus outbreak interventions. However, I find the introduction and discussion lacking in references of mathematical epidemiological models previously published (some e.g., PMID: 29084216, PMID: 25414312, PMID: 27383118, PMID: 28193880, PMID: 25358040). Several mathematical models have studied the role of intervention strategies and the distance of transmission. Whilst these do not take into account genome sequences and therefore do not have inferred historical events or inferred

interconnectedness of different locations, they mainly agree with the findings in this manuscript and therefore should be discussed with respect to the conclusions the authors make.

Some of the figures lack clarity and clear captions:

Figure 1A and B: what does the solid line correspond to?

Figure 1C and D: what do the brown and white bars correspond to? The authors mention a dashed line in Figure 1C but it is not clear what this refers to.

Additionally, the authors should clarify the statement "human mobility may not always be adequately modelled by a Brownian diffusion process" and provide a relevant reference.

The methods should be made clearer keeping in mind the need for reproducibility by independent researchers. For example, which tool was used to prune the trees, what program was used for re-estimating the effective population sizes with the Bayesian skygrid model. Additionally, the authors should consider making the XML files used for the continuous phylogeographic inference available. This would presumably contain the random coordinates assigned to the sequences with uncertain coordinates otherwise the analyses would not be directly reproducible. Are all the necessary files provided for a researcher to be able to reproduce the trees with 1607, 1567 [...] and 53 sequences (Page 12, line 293) and the final datasets of 722, 676 and 527 sequences (Page 13, line 329)?

It is commendable that the authors have provided a tutorial for the use of the posterior predictive simulation in the Seraphim package.

Finally, the references should be checked carefully as there are several typos.

Reviewer #3 (Remarks to the Author):

This study presents an extensive phylodynamic analysis of the 2013-2014 West Africa EBOV outbreak aimed at assessing the potential impact of intervention strategies on the epidemic spread.

This is an important work that addresses, in an indirect way, a highly debated topic in public health, i.e. the use of travel restrictions to contain epidemics. Such topic has been extensively studied in the past, typically with the use of computational or mathematical epidemic models based on human mobility and in the case of pandemic influenza.

Here, the authors present a novel approach to address the problem, by phylodynamic inference of spatial transmission events and provide novel and important insights into the spatial dynamics of the EBOV epidemic.

In particular, the results shed light on the role of large population centers in the EBOV spread and how preventing the infection from reaching the large cities in the affected area would have strongly reduced the burden of the epidemic.

Overall, this is a solid and high-quality research work based on state-of-the-art phylodynamic inference techniques. The results have important public health implications

and offer a new perspective for the assessment of intervention strategies in epidemic outbreaks.

Given the importance of the topic and the quality of the research, I believe this paper can be published in Nature Communications without the need of additional revisions.

I have only some minor remarks on the interpretation of the results and their implications for policy making which could be further discussed by the authors.

More specifically:

- the paper shows that by preventing the infection from reaching the major urban areas of the region would have curbed the number of new cases. In practical terms, would have it been feasible? What type of travel reduction would have been necessary to achieve such goal? Previous studies (such as Bajardi et al. PLOS ONE 2012, or Hollingsworth et al. Nat Med 2006) have shown that travel restrictions may be effective but practically unfeasible to contain the international spread of a pandemic. Can we derive some conclusion on this matter for the regional spread of EBOV in 2013-2014?

- if I understand correctly, the method presented by the authors allows to assess the impact of containment strategies only retrospectively, as it needs to rely on a large collection of genomic data from the outbreak. Can a similar analysis provide insights into the impact of border closure/travel restrictions during the outbreak, as done with computational epidemic models? Can it be used to assist policy makers in real-time to take decisions on intervention strategies during the outbreak?

Review 1:

This paper gives an account of a suite of phylodynamic/phylogeographic analyses of the recent West African Ebola outbreak. Although the sequences analysed have been published previously, by building on the results described in the previous paper this work generates important insights into the factors underpinning the outbreak. Specifically, the authors have used advanced phylogenetic approaches to evaluate the impact of long distance transmission, transmission to locations of defined population sizes, and the impact of border closures on the Ebola epidemic, as well as characterising the spatial spread of the virus. I find the paper to be well written, and interesting in both its methodological approaches and applied results. I feel the methods are appropriate and valid, and presented in sufficient detail to be reproducible. However, I do have some comments and queries on the manuscript which I feel should be addressed prior to publication:

Abstract

1. L9: Please clarify what is meant by “single locations”.

Answer: We have changed this to “movement to any given administrative area”.

2. L11: Please clarify this sentence with regards “transit centres’ for transmission chains”.

Answer: We have changed this to “urban areas ... were critical in attracting and further disseminating the virus”.

Introduction

3. Although the introduction provides useful background to the general literature surrounding the Ebola outbreak, it could be improved by refining the section to more specifically frame the work presented here and the questions it answers. This especially relates to the first paragraph of the Introduction.

Answer: We agree with the Reviewer and we have now reworked the first paragraph of the introduction. We now refer to containment efforts, how they have been investigated using mathematical modelling, and how sequence data and phylodynamic approaches may complement these to evaluate long-range interventions.

Results

4. L102-103: please could you be more clear about what the 97%/75% figure relates to, and ideally rewrite the sentence more clearly.

Answer: We have clarified this as follows: “...a 75% epidemic size reduction at 100 km instead of a 97% reduction without this restriction, Fig. 1A”.

5. L197: Is “cfr” a typo?

Answer: We have removed this abbreviation.

Materials and methods

6. L290 on: How well supported were the ancestral traits for the nodes of the MCC tree? If nodes with poor posterior trait support were present, how did were they accounted for in your analysis?

Answer: We have checked the support and more than 66% of the daughter nodes of a branch associated with a location transition have an ancestral state probability higher than 0.95. In addition, we have followed this up with further analyses that we detail in our answer to the next comment.

7. L290 on: Can you comment on whether pruning based on the subtrees identified from the MCC tree (rather than identifying the relevant subtrees individually for each of the posterior trees in turn) is likely to have influenced your results, or the confidence reported for your results?

Answer: We had to remove exactly the same taxa across all trees in the posterior so that we could also use the posterior genealogies to infer the demographic trajectories (our proxy of viral effective size through time). We would not be able to do this coalescent inference while averaging over empirical trees that contain different taxa. This is now explicitly mentioned in the Methods section. In addition, we have now examined how sensitive the tree height and tree length summaries are to conditioning on the MCC tree. We report the results in the Supplementary Information in the form of two new figures:

one that includes credible intervals for reductions in epidemic size and duration and one that is based on pruning that does not condition on the MCC tree. For the less important epidemic duration measure, we noticed that the reduction associated with preventing spread to administrative locations with >1,000k people is highly uncertain and not represented well by the MCC tree. We also highlight this in the main manuscript and thank the Reviewer for encouraging us to examine this.

8. L305-306: Can you comment on your rationale for using exclusive population size intervals here (e.g. 100-150k, 150-200k), as compared to the inclusive categories used for the distance-based pruning (L292-293).

Answer: Unlike for the distance-based pruning, where we can show a gradual effect on reduction in epidemic size (and duration), using cumulative population sizes would not be so informative. The reason is that for the category with the largest population sizes – the three capitals – we already see a reduction in epidemic size to about one third. For lower cut-offs in cumulative population size, this would obfuscate the large impact of Kenema and Kailahun districts (without delay in intervention). When applying a delay in intervention, we would only be able to show a marginal additional reduction (if any) for other locations if the capitals already account for a dramatic reduction in epidemic size.

9. Paragraph L330 on:

9a. I feel the location uncertainty in the data with respect to this part of the analysis should be better highlighted in the Results (e.g. at L168) and/or Discussion (e.g. L247-251).

Answer: We agree and this aspect is now better highlighted with the addition of the new analyses presented in Supplementary Information (see our answer to the comment “9c” and “10” below).

9b. Please provide a map of the administrative regions concerned in the supplement, so that the reader can get a better idea of the spatial resolution/uncertainty present in the data.

Answer: We have now included this as part of Figure 2B.

9c. Were the randomised location coordinates drawn separately at the start of each BEAST run, and, if so, did this lead to any differences in the estimates resulting from each independent run (if you ran multiple independent chains for each distance class, see point below)?

Answer: We fully address this comment in our reply to the next comment that is related to the issue raised here.

10. L350: Does this line indicate you ran only a single MCMC chain for each of the distance classes? If so, is there any concern over whether the single set of randomly-chosen starting locations are likely to affect the results for each of these distance classes (see the previous point)?

Answer: Random coordinates within an administrative area were drawn once prior to the BEAST analysis. Independent runs show that our phylogeographic estimates (e.g. dispersal velocity) are robust to the random coordinate draws. However, the Reviewer’s comment made us contemplate about the issue that the use of only one representative sequence (with randomly drawn coordinates) for a cluster of sequences specific to an administrative area may ignore many short-distance transmissions. To investigate to what extent this would affect our dispersal rate estimate, we have now analysed a data set of sequences for which more precise geographic coordinates were available and for which we did not restrict monophyletic clusters of sequences from the same administrative area to a single representative sequence. While this data set therefore accommodates transmission within administrative areas, and also differs in the time interval and total area of sampling (Sierra Leone), we arrive at remarkable consistent dispersal velocity estimates. This offers reasonable reassurance that our procedure does not result in strong biases. This additional analysis and related results are now included as Supplementary Information.

11. L369: Change “African” to “Africa”

Answer: Corrected.

Figures

12. Figure 1A and B: Please described in the figure legend the meaning of each of the lines (solid and dashed) and the shaded distribution. Additionally, it would be good to remove the (solid grey-ish) line from the edge of the distribution for clarity.

Answer: We have now completed the legend. The solid line was added for esthetical reasons because the colour gradient of the distribution plot is made of a collection of tiny polygons. We have made this line thinner for clarity.

13. Figure 1 C and D: No sign of the dashed line described in the legend. Please also indicate the meaning of the shaded and white bars.

Answer: Thank you for pointing out this oversight; the legend has now been corrected.

14. Figure 1 E: Might be best to change the colours to avoid green and red, in case of colour-blindness.

Answer: Thank you for this suggestion; we have now replaced the green colour by a light blue and checked the final result on <http://www.color-blindness.com>.

15. Figure 1 F contains a lot of information: is there any way of changing the figure to make this information any clearer or more accessible to the reader?

Answer: We attempted to optimise this figure in many different ways and tried to make a clearer distinction between the different curves by changing the position of the annotations.

16. Figure 4: Please indicate in the legend the meaning of the different colours. Again, please try and avoid green and red (for the colour-blind).

Answer: The figure and its legend have now been modified.

Review 2:

“Phylogenetic assessment of intervention strategies for West African Ebola virus outbreak” by Dellicour et al. presents some very interesting insight into the implications of different intervention scenarios on the spread and size of the Ebola virus outbreak. The authors use phylogenetic approaches to explore different intervention strategies. They show that 1) long distance dispersal events did not have a large effect on the epidemic expansion, 2) that the three capitals were hubs of transmission and isolating these at the same time would only have reduced the epidemic size by one third, 3) that the dispersal velocity increased steeply from January 2014 to June and decreased markedly following border closures in July 2014 and this pattern was still present in the absence of branches crossing borders suggesting that the decrease in dispersal velocity could be attributed to general awareness of the outbreak rather than border closure.

This manuscript is novel and of broad importance to the research community, additionally it is likely to influence future Ebola virus outbreak interventions. However, I find the introduction and discussion lacking in references of mathematical epidemiological models previously published (some e.g., PMID: 29084216, PMID: 25414312, PMID: 27383118, PMID: 28193880, PMID: 25358040). Several mathematical models have studied the role of intervention strategies and the distance of transmission. Whilst these do not take into account genome sequences and therefore do not have inferred historical events or inferred interconnectedness of different locations, they mainly agree with the findings in this manuscript and therefore should be discussed with respect to the conclusions the authors make.

Answer: We thank the Reviewer for pointing this out. We have now discussed many of these modelling studies in the introduction/discussion.

Some of the figures lack clarity and clear captions:

Figure 1A and B: what does the solid line correspond to?

Figure 1C and D: what do the brown and white bars correspond to? The authors mention a dashed line in Figure 1C but it is not clear what this refers to.

Answer: We apologize for this oversight. The figure legends have been completed and corrected.

Additionally, the authors should clarify the statement “human mobility may not always be adequately modelled by a Brownian diffusion process” and provide a relevant reference.

Answer: We have modified the phrasing and we now refer to Faria *et al.* (2011, DOI 10.1016/j.coviro.2011.10.003) where this is discussed in more detail.

The methods should be made clearer keeping in mind the need for reproducibility by independent researchers. For example, which tool was used to prune the trees, what program was used for re-estimating the effective population sizes with the Bayesian skygrid model. Additionally, the authors should consider making the XML files used for the continuous phylogeographic inference available. This would presumably contain the random coordinates assigned to the sequences with uncertain coordinates otherwise the analyses would not be directly reproducible. Are all the necessary files provided for a researcher to be able to reproduce the trees with 1607, 1567 [...] and 53 sequences (Page 12, line 293) and the final datasets of 722, 676 and 527 sequences (Page 13, line 329)?

Answer: All the XML files are now available online (<https://github/ebov/space-time/data>). We now also provide more detail about the tools used to prune the trees and to estimate the effective population sizes.

It is commendable that the authors have provided a tutorial for the use of the posterior predictive simulation in the Seraphim package.

Answer: Thank you for appreciating our efforts.

Finally, the references should be checked carefully as there are several typos.

Answer: We have checked and corrected all the references.

Review 3:

This study presents an extensive phylodynamic analysis of the 2013-2014 West Africa EBOV outbreak aimed at assessing the potential impact of intervention strategies on the epidemic spread.

This is an important work that addresses, in an indirect way, a highly debated topic in public health, i.e. the use of travel restrictions to contain epidemics. Such topic has been extensively studied in the past, typically with the use of computational or mathematical epidemic models based on human mobility and in the case of pandemic influenza. Here, the authors present a novel approach to address the problem, by phylodynamic inference of spatial transmission events and provide novel and important insights into the spatial dynamics of the EBOV epidemic. In particular, the results shed light on the role of large population centers in the EBOV spread and how preventing the infection from reaching the large cities in the affected area would have strongly reduced the burden of the epidemic. Overall, this is a solid and high-quality research work based on state-of-the-art phylodynamic inference techniques. The results have important public health implications and offer a new perspective for the assessment of intervention strategies in epidemic outbreaks. Given the importance of the topic and the quality of the research, I believe this paper can be published in Nature Communications without the need of additional revisions.

I have only some minor remarks on the interpretation of the results and their implications for policy making which could be further discussed by the authors.

More specifically:

- the paper shows that by preventing the infection from reaching the major urban areas of the region would have curbed the number of new cases. In practical terms, would have it been feasible? What type of travel reduction would have been necessary to achieve such goal? Previous studies (such as Bajardi *et al.* PLOS ONE 2012, or Hollingsworth *et al.* Nat Med 2006) have shown that travel restrictions may be effective but practically unfeasible to contain the international spread of a pandemic. Can we derive some conclusion on this matter for the regional spread of EBOV in 2013-2014?

Answer: The Reviewer raises an important point and we had not addressed this in the discussion of the previous version of our manuscript. We now clearly emphasize that 1) the intervention measures we assess represent a best-case scenario as they are assumed to be 100% successful, and that 2) effectively realizing these interventions would be complicated in practice (and probably impossible if they need to be 100% successful). Yet, we believe that reporting the findings related to this best-case scenario remains important as they capture important aspects, such as the critical role of the capitals, and also measures that are not 100% successful may prove useful given the metapopulation dynamics that were observed for EBOV spread.

- if I understand correctly, the method presented by the authors allows to assess the impact of containment strategies only retrospectively, as it needs to rely on a large collection of genomic data from the outbreak. Can a similar analysis provide insights into the impact of border closure/travel restrictions during the outbreak, as done with computational epidemic models? Can it be used to assist policy makers in real-time to take decisions on intervention strategies during the outbreak?

Answer: Our approach indeed reconstructs the history of viral spread up to the most recent time of sampling, and therefore remains limited in its use for predictive purposes. It may however elucidate the frequency of cross-border spread, and by using this information in computational models, it may assist in predicting the magnitude of the impact of border closures. We did not dwell on this issue as we believe the primary 'real-time' contribution of molecular sequence analyses in practice will be transmission chain reconstruction and assisting contact tracing efforts.

REVIEWERS' COMMENTS:

Reviewer #1 (Remarks to the Author):

I thank the authors for their response to my review. I am happy with their changes and have no further comments on the manuscript.